# Real-Time PCR Method for the Rapid Detection and Quantification of Pathogenic *Staphylococcus* Species Based on Novel Molecular Target Genes

**DOI:** 10.3390/foods10112839

**Published:** 2021-11-17

**Authors:** Eiseul Kim, Seung-Min Yang, Ji-Eun Won, Da-Young Kim, Da-Som Kim, Hae-Yeong Kim

**Affiliations:** Institute of Life Sciences & Resources, Department of Food Science and Biotechnology, Kyung Hee University, Yongin 17104, Korea; eskim89@khu.ac.kr (E.K.); ysm9284@gmail.com (S.-M.Y.); wldms7183@naver.com (J.-E.W.); ekdudvv4589@naver.com (D.-Y.K.); dasove22@naver.com (D.-S.K.)

**Keywords:** *Staphylococcus aureus*, *Staphylococcus capitis*, *Staphylococcus caprae*, *Staphylococcus epidermidis*, real-time PCR, pan-genome analysis, detection

## Abstract

Coagulase-positive *Staphylococcus aureus* is a foodborne pathogen considered one of the causes of food-related disease outbreaks. Like *S. aureus*, *Staphylococcus capitis*, *Staphylococcus caprae*, and *S. epidermidis* are opportunistic pathogens causing clinical infections and food contamination. The objective of our study was to develop a rapid, accurate, and monitoring technique to detect four *Staphylococcus* species in food. Four novel molecular targets (GntR family transcriptional regulator for *S. aureus*, phosphomannomutase for *S. epidermidis*, FAD-dependent urate hydroxylase for *S. capitis*, and Gram-positive signal peptide protein for *S. caprae*) were mined based on pan-genome analysis. Primers targeting molecular target genes showed 100% specificity for 100 non-target reference strains. The detection limit in pure cultures and artificially contaminated food samples was 10^2^ colony-forming unit/mL for *S. aureus*, *S. capitis*, *S. caprae*, and *S. epidermidis*. Moreover, real-time polymerase chain reaction successfully detected strains isolated from various food matrices. Thus, our method allows an accurate and rapid monitoring of *Staphylococcus* species and may help control staphylococcal contamination of food.

## 1. Introduction

Staphylococci are common pathogens, widely distributed in nature and frequently isolated from food. *Staphylococcus aureus* is one of the most important foodborne pathogens, producing staphylococcal enterotoxins causing diarrhea and vomiting through direct invasion or systemic transmission, adversely affecting human health [1,2]. Unlike foodborne pathogenic *S. aureus*, which is a coagulase-positive *Staphylococcus*, *S. capitis*, *S. caprae*, and *S. epidermidis* are coagulase-negative staphylococci (CoNS) [3]. Traditionally considered commensals, CoNS species are now recognized as opportunistic pathogens [4]. Moreover, CoNS species contaminate foods and have become a prominent pathogenic strain in ready-to-eat foods [1]. In the CoNS group, *S. epidermidis* is the most common pathogen associated with human infections, such as bacteremia and endocarditis in immunocompromised patients [5]. *S. capitis* is occasionally associated with hospital-acquired meningitis and native and prosthetic valve endocarditis. At the same time, *S. caprae* infections include bacteremia and prosthetic infections [4,5,6]. *S. saccharolyticus* has been reported to be associated with human infection but has rarely been isolated from food. Phylogenetic analysis using the 16S rRNA gene classified *S. epidermidis* into the *S. epidermidis* group, including *S. capitis*, *S. caprae*, and *S. saccharolyticus* [5]. These species are closely phenotypically related, so it is difficult to distinguish them. Therefore, accurate methods to detect and discriminate *Staphylococcus* species contamination in food are needed to reduce disease outbreaks and ensure food safety.

Traditional methods for detecting pathogenic bacteria require multiple steps, including pre-enrichment, selective growth, and selective plating, which can then be characterized by analyzing additional biochemical tests [7]. The process is labor-intensive and time-consuming, is not cost-effective, and is also unable to detect viable but nonculturable cells [8,9]. Various staphylococci detection methods exist, both phenotypic and genotypic. Other detection methods, such as the Staph-Zym test, the API Staph test, and the BF Phenix Automated Microbiology system, have been used for the detection of staphylococci based on their phenotypic properties [10,11]. Recently, an attempt has been made to detect staphylococci using matrix-assisted laser desorption ionization–time-of-flight mass spectrometry based on protein profiles [12]. However, the low accuracy (50–70%) of the biochemical reaction of the API kit and the high initial acquisition cost of automatic mass spectrometry systems restrict their application [13,14]. Compared to genotypic detection methods, such as amplified fragment length polymorphic fingerprinting and polymerase chain reaction (PCR)-based methods, phenotypic tests are less accurate [1,14,15].

The development of simple and rapid methods with high specificity and sensitivity is critical for detecting pathogenic bacteria. In recent decades, molecular detection methods, such as PCR, have been widely employed for pathogen detection because they are faster and simpler than conventional culture methods [16,17]. Among the PCR-based methods, real-time PCR has become a useful tool for detecting and quantifying bacterial species associated with foods due to its superiority, such as high sensitivity and efficiency [8,17,18,19,20,21]. At present, real-time PCR is used for monitoring staphylococci in food processing [1,22,23].

The selection of appropriate genes or sequences is critical for pathogenic bacteria detection using PCR. Various molecular target genes were used to detect *Staphylococcus* species, namely, 16S rRNA gene, *tuf* (elongation factor Tu), *sodA* (superoxide dismutase A), *nuc* (thermostable nuclease), and *dnaJ* (chaperone dnaJ) [11,24,25,26]. However, some of these genes cannot discriminate between phylogenetically closely related species, such as the *S. epidermidis* and *S. aureus* cluster groups. The lack of exclusivity and inclusivity of molecular target genes is one major drawback of implementing the PCR-based method in food inspection [27]. Advances in high-throughput sequencing technology have allowed increasing the number of whole-genome sequences available in public databases, making it easier to obtain target genes specific for *Staphylococcus* species using bioinformatics approaches [27]. This study aimed to develop a real-time PCR assay targeting molecular target genes to detect the staphylococci *S. aureus*, *S. capitis*, *S. caprae*, and *S. epidermidis*, thus allowing their accurate and rapid monitoring in food matrix.

## 2. Materials and Methods

### 2.1. Evaluation of Staphylococci Genomes

The genome sequences of 155 *Staphylococcus* strains were obtained from the National Center for Biotechnology Information (NCBI). *Staphylococcus* genomes consist of the *S. aureus* cluster group (8 *S. argenteus*, 35 *S. aureus*, 9 *S. schweitzeri*, 9 *S. simiae*), the *S. epidermidis* cluster group (16 *S. capitis*, 15 *S. caprae*, 28 *S. epidermidis*, 18 *S. saccharolyticus*), 9 *S. pasteuri*, 8 *S. warneri*. Detailed genome information is shown in Appendix A. Phylogeny analysis was performed using anvi’o software version 7.0 [28] to confirm the taxonomic position of the genomes obtained from publicly available databases. Assembled genomes were first used to generate genome storage files using the “anvi-gene-genomes-storage” command. Genome storage was then used for phylogeny analysis based on pan-genome using the “anvi-pan-genome” command. The average nucleotide identity (ANI) value was calculated using the “anvi-compute-ani” command, which uses PyANI [29].

### 2.2. Mining of Molecular Target Genes

The molecular target genes were mined using bacterial pan-genome analysis (BPGA) v1.3 [30]. The criteria for selecting molecular target genes of four species were 100% presence in the respective target species and absence in non-target species. The genomes of staphylococci were constructed using two databases: a core gene database for target species and a pan gene database for non-target species. Then, the two databases were compared to search for target genes with a cut-off value of 50%, default parameter. The candidate target genes were searched using BLAST to further identify the genes specific to each species. The genes that were absent, with 72,899,005 other bacterial sequences, were considered molecular target genes. The specificity of the discovered molecular target genes and of the reported target genes (*tuf*, *sodA*, *nuc*, and *dnaJ*) was confirmed by aligning them with 94 *S. aureus*, *S. capitis*, *S. caprae*, and *S. epidermidis* genomes.

### 2.3. Design of Specific Primers

Based on the sequences of the target genes, primer pairs were designed using Primer Designer (Scientific and Education Software, Durham, NC, USA) with the following criteria: G + C content, 40–60%, Tm value, 65 °C and 75 °C, and no ability to form dimers. The specificity of the primer pairs was checked using the primer-BLAST tool [31].

### 2.4. DNA Extraction

The reference strains, including 39 *Staphylococcus* strains and 73 non-*Staphylococcus* strains, are listed in Table 1. All reference strains were grown in tryptic soy broth (TSB, Difco, Becton & Dickinson, Sparks, MD, USA) at 37 °C for 24 h. Genomic DNA of all staphylococci and non-staphylococci strains was extracted using the G-spin genomic DNA extraction kit (Intron Biotechnology, Seongnam, Korea) according to the manufacturer’s instructions. The DNA purity and concentration were measured using a NanoDrop spectrophotometer (Thermo Fisher Scientific, Waltham, MA, USA).

### 2.5. Real-Time PCR Condition

Real-time PCR was performed in a 20 µL reaction mixture containing 10 µL of 2× Thunderbird SYBR^®^ qPCR mix (Toyobo, Osaka, Japan), 1 µL of each primer (10 pmol/µL), 1 µL of template (20 ng/µL), and 7 µL of distilled water. A CFX96 Touch Deep (Bio-Rad, Hercules, CA, USA) was used for thermal cycling as follows: denaturation at 95 °C for 5 min, followed by 35 cycles at 95 °C for 5 s and 60 °C for 30 s. The melting curve was obtained by increasing the temperature between 60 °C and 95 °C in 0.5 °C increments while holding for 15 s at each step.

### 2.6. Specificity of the Primer Pairs

The specificity of the primer pairs was confirmed by 112 bacterial strains (Table 1). If a primer pair successfully produced an amplification plot from strains of the corresponding species, it was then tested against other *Staphylococcus* strains and non-*Staphylococcus* strains. For sensitivity testing, cultures of *S. aureus* ATCC 6538, *S. epidermidis* KACC 13234, *S. capitis* KACC 13242, and *S. caprae* KCTC 3583 were serially diluted (10^2^–10^8^ colony-forming units (CFU/mL), and genomic DNA was extracted as described in Section 2.4, followed by real-time PCR.

### 2.7. Application in Artificially Contaminated Food Samples

The reliability of the established real-time PCR assay for pathogenic *Staphylococcus* detection in the food matrix was determined according to previous studies [1,19]. All strains were grown in TSB for 24 h at 37 °C, and plate counting was performed in TSA medium. For the spiking experiment, milk samples were purchased, and the absence of four *Staphylococcus* species was confirmed. Spiked samples were prepared by inoculating the cocktail of *S. aureus* ATCC 6538, *S. epidermidis* KACC 13234, *S. capitis* KACC 13242, and *S. caprae* KCTC 3583 at a concentration of 10^2^–10^8^ CFU/mL each in 25 mL of milk samples. The prepared spiked samples were then homogenized for 1 min. The genomic DNA of each homogenized sample was extracted under the conditions described in Section 2.4. Real-time PCR analysis was conducted at the conditions described in Section 2.5.

### 2.8. Application in Isolated Staphylococcus Strains

Twenty-two samples (4 chicken, 3 beef, 5 pork, 3 fish, 5 salted fish, and 2 raw milk) were collected in Korea to confirm the feasibility of the detection of staphylococci by real-time PCR. Twenty-five grams of food sample was transferred to stomacher bags and then homogenized with 225 mL saline and mixed thoroughly to isolate *Staphylococcus* strains. Serial dilutions of the homogenized samples were prepared, and a small volume of each of them was spread on mannitol salt agar (Difco) media and incubated at 37 °C for 24 h. Then, the genomic DNAs of unknown isolates were used for identification by the real-time PCR method developed in this study.

### 2.9. Detection of Staphylococcus Species in Food Samples

Fifty samples, including 2 samples of ready-to-eat vegetables (1 cucumber and 1 lettuce), 10 samples of meat (2 beef and 8 pork), 34 samples of raw milk, and 4 samples of salted fish, were randomly collected from local markets in Korea. Ten grams of each sample was homogenized using a blender. The genomic DNA of the food samples was extracted under the conditions described in Section 2.4. Detection of four *Staphylococcus* species in unknown samples was performed under the conditions described in Section 2.5.

## 3. Results

### 3.1. Evaluation of Staphylococci Whole-Genome Sequences

Misclassified genomes for closely related bacteria are often reported in public database [20,32,33]. The phylogenetic tree based on pan-genome cluster frequencies was clustered according to the species name, and two clusters were generated (Figure 1). The first cluster included *S. aureus*, *S. argenteus*, and *S. schweitzeri*, and the second cluster included *S. epidermidis*, *S. simiae*, *S. warneri*, *S. pasteuri*, *S. saccharolyticus*, *S. caprae*, and *S. capitis*. However, some genomes, such as those of *S. pasteuri* SP1, *S. saccharolyticus* IIF7SG_B1, 151250007-1-258-46, and IIF6SC-B4A, clustered with those of other species. *S. pasteuri* SP1 clustered with *S. warneri*. *S. saccharolyticus* IIF7SG_B1, 151250007-1-258-46, and IIF6SC-B4A clustered with *S. pasteuri*. In OrthoANI analysis, *S. pasteuri* SP1 showed 95.95% identity with an *S. warneri*-type strain (NCTC 11044) and 83.37% identity with an *S. pasteuri*-type strain (DSM 10656). In addition, *S. saccharolyticus* IIF7SG_B1, 151250007-1-258-46, and IIF6SC-B4A showed 98.93–98.98% identity with an *S. pasteuri*-type strain (DSM 10656) and 76.72–76.95% identity with an *S. saccharolyticus*-type strain (NCTC 11807). These results suggest that these strains should be corrected in the public database to avoid further misidentification.

### 3.2. Pan-Genome Analysis

One hundred fifty-five genomes were retrieved for pan-genome analysis. To screen target genes for detecting the four *Staphylococcus* species, 388,852 coding genes from 155 staphylococci genomes yielded a pan-genome of 9461 genes consisting of 464 core-genes, 6772 accessory-genes, and 2225 unique-genes. Of the 6772 accessory genes, 4, 13, 19, and 18 genes were common to 35 *S. aureus*, 16 to *S. capitis* genomes, 15 to *S. caprae*, and 28 to *S. epidermidis*. Fifty-four candidate genes were aligned with other sequences through the blast program. The molecular target genes were mined based on their 100% presence in all genomes of the target species and their absence in the genomes of non-target staphylococci. The functions of the molecular target genes selected for the detection of *Staphylococcus* species are as follows: GntR family transcriptional regulator (accession no. ABD29409.1) for *S. aureus*, FAD-dependent urate hydroxylase (accession no. AKL93396.1) for *S. capitis*, Gram-positive signal peptide protein, YSIRK family (accession no. BBD88784.1) for *S. caprae*, and phosphomannomutase (accession no. AAO05683.1) for *S. epidermidis*.

The specificity of the discovered molecular target genes in comparison with that of the previously reported target genes in detecting four *Staphylococcus* species was confirmed; we observed higher specificity for our molecular target genes (Appendix A). All target genes except the *nuc* gene were present with high similarities (86–100%) not only in the target species but also in the non-target species. These results indicated that the four molecular target genes are suitable for the identification of *Staphylococcus* species.

Specific primers were designed for each molecular target gene, and the specificity was confirmed through in silico analysis. As a result, all primer pairs produced amplicons only in the target species (Appendix A). For *S. aureus*-specific primers, we found that certain genome sequences, including those of *Staphylococcus* species MZ1, MZ3, MZ7, MZ8, MZ9, T93, SM3655, and SM9054, produced an amplicon of 145 bp. Eight genomes of *Staphylococcus* species showed 98.83 to 99.87% identity with an *S. aureus*-type strain (DSM 20231), suggesting that these genomes belong to *S. aureus*.

### 3.3. Specificity and Sensitivity of Real-Time PCR

A real-time PCR was designed to identify the four staphylococci species that cause food poisoning and are often isolated from food [34]. The specificity was tested using 12 target *Staphylococcus* strains, 27 non-target *Staphylococcus* strains, and 73 non-*Staphylococcus* strains (Table 1). The primer sequences and amplicon size are shown in Table 2. All *S. aureus*, *S. capitis*, *S. caprae*, and *S. epidermidis* strains yielded detectable amplicons for the corresponding primer pairs, whereas no amplifications were generated with the non-target *Staphylococcus* strains and non-*Staphylococcus* strains, indicating the high specificity of the four primer pairs (Figure 2).

The sensitivity analysis was performed using DNA from serial dilutions of the target bacterial species (10^2^–10^8^ CFU/mL) as a template. All tests were repeated thrice, and the standard curves are presented in Figure 3. For the detection of *S. aureus*, the sensitivity was 1.5 × 10^2^ CFU/mL. Similarly, the limit of detection of *S. capitis*, *S. caprae*, and *S. epidermidis* was 2.6 × 10^2^ CFU/mL, 1.4 × 10^2^ CFU/mL, and 1.09 × 10^2^ CFU/mL, respectively. The *R*^2^ values were higher than 0.997, indicating that the amounts of DNA showed high linearity with the corresponding Ct values [35]. The equations for *S. aureus*, *S. capitis*, *S. caprae*, and *S. epidermidis* were y = −3.582x + 41.759, y = −3.293x + 39.33, y = −3.522x + 41.263, and y = −3.134x + 40.753 respectively. The amplification efficiencies for the four *Staphylococcus* species ranged from 90.20% to 108.50%, indicating high efficiency [35].

### 3.4. Detection of Staphylococcus Species in Artificially Contaminated Food Samples

DNA present in a food matrix can impair the efficiency of real-time PCR, as its concentration can be underestimated [36]. In this study, the quantification in food matrix was conducted by artificially adding four *Staphylococcus* species to milk, a food product they mainly inhabit. Simultaneously, the milk samples were inoculated with a cocktail of four pathogenic *Staphylococcus* species to induce competition between strains for the same nutrients. Food samples artificially inoculated with *Staphylococcus* species (average Ct values: 14.11–33.27) had Ct values similar to those of pure cultured bacteria (average Ct values: 13.7–33.82). All standard curves showed high efficiency, with *R^2^* of 0.998 for *S. aureus*, 0.997 for *S. capitis* and *S. epidermidis*, and 0.994 *S. caprae* (Figure 4). The limit of detection values were 1.5 × 10^2^ CFU/mL for *S. aureus*, 2.6 × 10^2^ CFU/mL for *S. capitis*, 1.4 × 10^2^ CFU/mL for *S. caprae*, and 1.2 × 10^2^ CFU/mL for *S. epidermidis*. The detection limits for the four *Staphylococcus* species in artificially contaminated milk samples were similar to those in pure cultures. These results suggested that the real-time PCR method could detect the four *Staphylococcus* species almost without any interference from the food matrix.

### 3.5. Real-Time PCR Detection of Isolates

A total of 103 strains were isolated from chicken, beef, pork, fish, salted fish, and raw milk. All isolates produced one amplification curve: 36 isolates (34.95%), 63 isolates (61.17%), and 4 isolates (3.88%) were identified as *S. aureus*, *S. epidermidis*, and *S. capitis* (Table 3). *S. aureus* was isolated from meat such as chicken, beef, pork, and fish. *S. epidermidis* was isolated from pork, salted fish, and raw milk. *S. capitis* was isolated only from salted fishes, and *S. caprae* was not isolated from any sample.

### 3.6. Detection of Contamination by the Four Staphylococcus Species in Food Samples

To confirm the efficacy of real-time PCR for the detection of *Staphylococcus* species contamination in food samples, 50 samples were tested using the real-time PCR method developed in this study. *S. aureus* was detected in 11 samples of pork and raw milk, and *S. epidermidis* was detected in 9 samples of raw milk. *S. capitis* was detected in three samples of fermented fish and raw milk. *S. caprae* was not present in any of the food samples. The result showed that the detection rates of the *Staphylococcus* species were 22% for *S. aureus*, 6% for *S. capitis*, and 18% for *S. epidermidis* (Table 4). These results are consistent with a previous study suggesting that *S. epidermidis* and *S. aureus* are the main *Staphylococcus* species contaminating food [1].

## 4. Discussion

*S. aureus* and *S. epidermidis* are important pathogenic bacteria that cause clinical infections and food contamination [7,37]. These two pathogens are isolated in a wide range of foods, such as vegetables, meat, and fish [38]. *S. capitis* and *S. caprae* are species closely related to *S. epidermidis*, an opportunistic CoNS. They contaminate milk or meat and have been isolated from fermented foods such as cheese [34]. Therefore, developing reliable and rapid methods to detect these four pathogenic *Staphylococcus* species has become increasingly important to protect public health and ensure food safety [17]. Here, we developed a rapid and accurate detection method for four pathogenic *Staphylococcus* species based on novel molecular target genes.

Molecular detection methods play an important role in rapidly detecting pathogenic bacteria [27]. The usefulness of molecular detection methods is dependent on the target genes or sequences and the specificity of specific primer pairs [27]. The current PCR methods for pathogenic staphylococci target 16S rRNA genes, housekeeping genes, or virulence genes [26,38]. However, a previous study has reported that the 16S rRNA genes of the *S. epidermidis* group share high sequence similarities (≥97%) and do not exhibit sufficient variability to allow differentiating the species [39]. In addition, the lack of virulence genes can result in misclassification, posing a potential threat of food poisoning [1,40]. As numerous whole-genome sequences become available with the development of genome sequencing technologies, many researchers are committed to the search for novel molecular target genes to replace the current markers that exhibit poor specificity [20,21,40,41]. In this study, pan-genome analysis was utilized for discovering molecular target genes of *Staphylococcus* species. We successfully identified molecular target genes specific for four *Staphylococcus* species via a pan-genome analysis. At the same time, we found misclassified staphylococci genomes. Through pan-genome analysis, we found that four molecular target genes were 100% specific for identifying *Staphylococcus* species and did not match other bacterial genes.

PCR methods are specific for detecting causative pathogens of infectious diseases and for discriminating closely related species [42]. Previous studies have reported that the PCR-based detection method of *Staphylococcus* species is more rapid, easier, and sensitive than traditional methods [1,7,11,43]. Real-time PCR methods provide a tool for the sensitive and accurate quantification of target bacteria, which could be applied to detect *Staphylococcus* species in foods [22,23,44]. Although several real-time PCR methods for detecting *Staphylococcus* species have been reported, their target genes or sequences have been shown poor specificity [1]. Recently, a real-time PCR method targeting novel specific genes obtained by a pan-genome analysis for the accurate detection of *Staphylococcus* species has been developed [1]. This method displayed a better specificity than the previous real-time PCR method. However, for monitoring *Staphylococcus* species using existing real-time PCR methods, previous studies focused on discovering novel genes for *S. aureus* and *S. epidermidis* [1], while no target genes for *S. capitis* and *S. caprae*, which are closely related species to *S. epidermidis*, have been reported. In this study, we successfully identified molecular target genes for *S. capitis* and *S. caprae* with high specificity and sensitivity. More surprisingly, the detection limit of *S. aureus* (10^2^ CFU/mL) was equivalent to that of previously reported target genes [1]. In contrast, the detection limit for *S. epidermidis* molecular target gene showed an obvious advantage in this study. The real-time PCR method developed in this study maintained a good consistency in detecting the four *Staphylococcus* species, without interference from the food matrix. Moreover, this method was successfully applied to 103 strains isolated from chicken, beef, pork, fish, salted fish, and raw milk. These results indicate that the molecular target genes discovered in this study have specificity in real-time PCR analysis, allowing the rapid, accurate, and sensitive detection of the four *Staphylococcus* species in food matrix.

In conclusion, we successfully mined four molecular target genes for the four *Staphylococcus* species *S. aureus*, *S. capitis*, *S. caprae*, and *S. epidermidis*. We developed a real-time PCR to detect the four *Staphylococcus* species with high specificity and high sensitivity. Our real-time PCR method was able to successfully detect the four pathogenic *Staphylococcus* species in food. Our data show that the method has a great potential as an accurate, rapid, and sensitive tool to monitor potential pathogenic *Staphylococcus* species in food samples.

## Figures and Tables

**Figure 1 foods-10-02839-f001:**
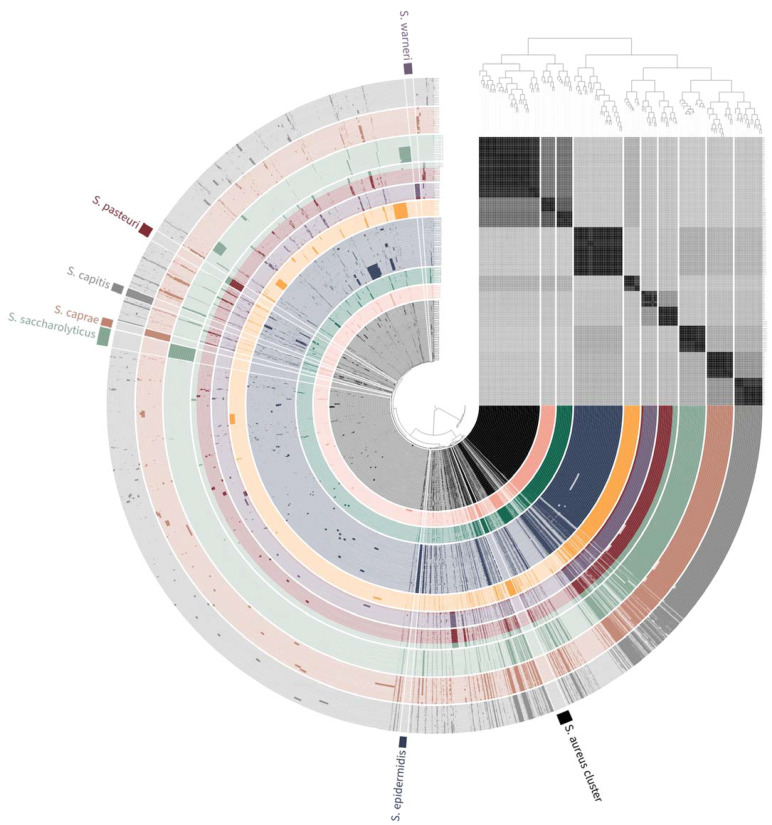
Pan-genome analysis generated with Anvi’o software (version 7.0) for 155 staphylococci genomes. The layers represent individual staphylococci genomes organized by their phylogenomic relationship. In the layers, the dark and bright areas within the bars indicate the presence and absence of genes, respectively. The ANI values are represented by a heatmap determined at high (black) and low (gray) similarities.

**Figure 2 foods-10-02839-f002:**
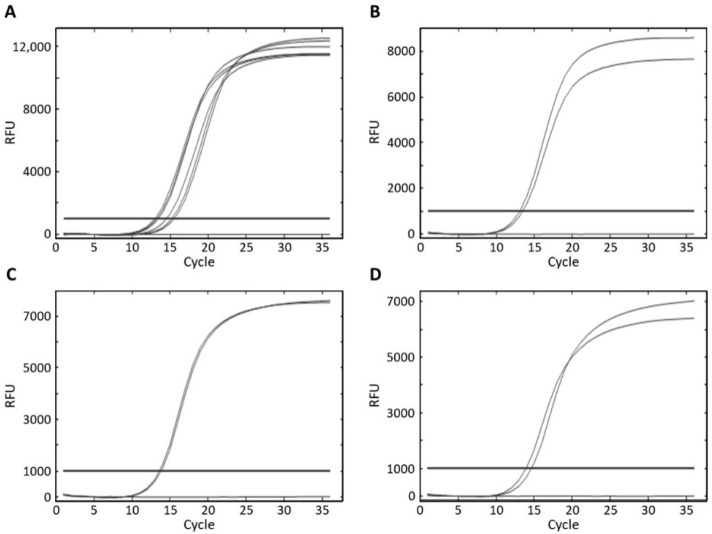
Specificity of the primer pairs for real-time PCR amplification. (**A**) *S. aureus* ATCC 29213, KCTC 1928, NCCP 14560, ATCC25923, ATCC 29737, and ATCC 6538 amplified using the *S. aureus* primer pair; (**B**) *S. capitis* NCCP 14663 and KACC 13242 amplified using the *S. capitis* primer pair; (**C**) *S. caprae* KCTC 3583 and NCCP 15629 amplified using the *S. caprae* primer pair; (**D**) *S. epidermidis* NCCP 14723 and KACC 13234 amplified using the *S. epidermidis* primer pair.

**Figure 3 foods-10-02839-f003:**
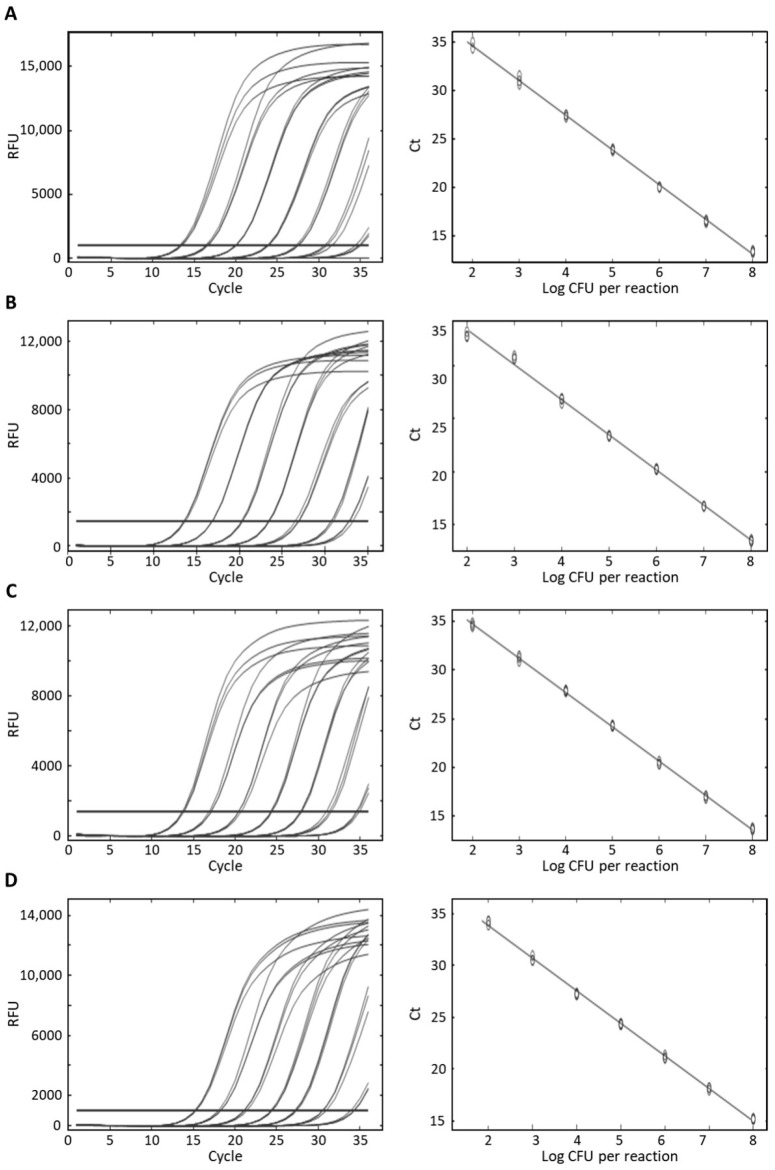
Standard curves by plotting cycle threshold (Ct) values against the logarithm of the number of cells of (**A**) *S. aureus*, (**B**) *S. capitis*, (**C**) *S. caprae*, and (**D**) *S. epidermidis* in pure culture.

**Figure 4 foods-10-02839-f004:**
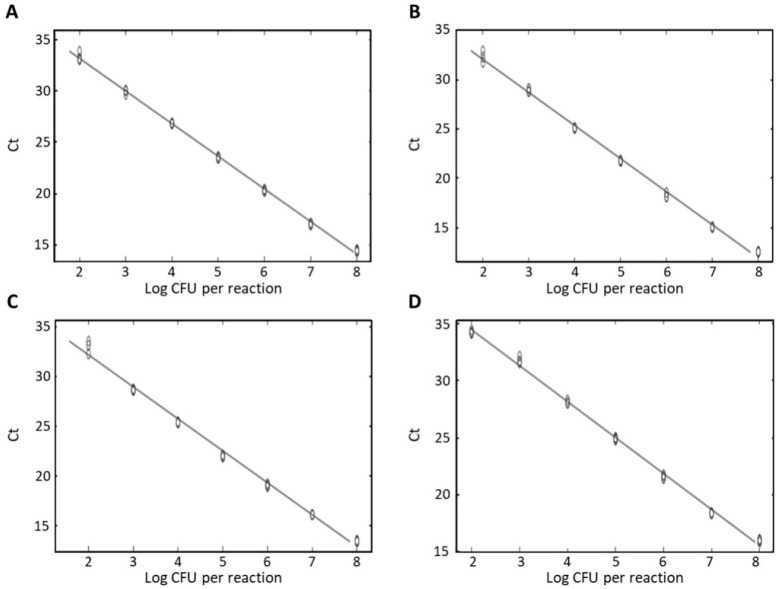
Limit of detection for (**A**) *S. aureus*, (**B**) *S. capitis*, (**C**) *S. caprae*, and (**D**) *S. epidermidis* in spiked milk samples.

**Table 1 foods-10-02839-t001:** List of reference strains used in this study.

Classification	Species	Strain Number
Target *Staphylococcus*	*Staphylococcus aureus*	ATCC 29213
	*Staphylococcus aureus*	KCTC 1928
	*Staphylococcus aureus*	NCCP 14560
	*Staphylococcus aureus*	ATCC 25923
	*Staphylococcus aureus*	ATCC 29737
	*Staphylococcus aureus* subsp. *aureus*	ATCC 6538
	*Staphylococcus capitis*	NCCP 14663
	*Staphylococcus capitis* subsp. *capitis*	KACC 13242
	*Staphylococcus caprae*	KCTC 3583
	*Staphylococcus caprae*	NCCP 15629
	*Staphylococcus epidermidis*	NCCP 14723
	*Staphylococcus epidermidis*	KACC 13234
Non-target *Staphylococcus*	*Staphylococcus auricularis*	KACC 13252
	*Staphylococcus carnosus* subsp. *utilis*	KACC 13190
	*Staphylococcus cohnii* subsp. *cohnii*	KACC 13237
	*Staphylococcus cohnii* subsp. *urealyticus*	KCTC 3574
	*Staphylococcus delphini*	KACC 13258
	*Staphylococcus equorum* subsp. *equorum*	KACC 13255
	*Staphylococcus fleurettii*	KACC 13199
	*Staphylococcus gallinarum*	KACC 13253
	*Staphylococcus haemolyticus*	NCCP 14691
	*Staphylococcus hominis*	NCCP 10748
	*Staphylococcus hominis*	KACC 13409
	*Staphylococcus kloosii*	KACC 13256
	*Staphylococcus lentus*	KCCM 41469
	*Staphylococcus lugdunensis*	NCCP 15630
	*Staphylococcus lugdunensis*	KACC 11270
	*Staphylococcus pasteuri*	KCTC 13167
	*Staphylococcus pettenkoferi*	DSM 19554
	*Staphylococcus saprophyticus*	NCCP 14670
	*Staphylococcus saprophyticus*	KCTC 3345
	*Staphylococcus saprophyticus*	KACC 15799
	*Staphylococcus schleiferi* subsp. *coagulans*	KCCM 41634
	*Staphylococcus sciuri* subsp. *rodentium*	KACC 13217
	*Staphylococcus sciuri* subsp. *sciuri*	KCCM 41468
	*Staphylococcus warneri*	KCTC 3340
	*Staphylococcus warneri*	KACC 10785
	*Staphylococcus xylosus*	NCCP 10937
	*Staphylococcus xylosus*	KACC 13239
Non-*Staphylococcus*	*Bacillus cereus*	KCTC 3624
	*Bacillus cereus*	KCTC 1661
	*Bacillus cereus*	KCCM 1173
	*Bacillus cereus*	KCCM 1174
	*Bacillus cereus*	KCCM 40133
	*Bacillus cereus*	ATCC 11778
	*Bacillus cereus*	ATCC 10876
	*Bacillus cereus*	ATCC 14579
	*Bacillus circulans*	KCTC 3347
	*Bacillus licheniformis*	KCTC 1026
	*Bacillus megaterium*	KCTC 3007
	*Bacillus subtilis*	KCTC 3725
	*Clostridium perfringens*	ATCC 14810
	*Enterococcus avium*	KACC 10788
	*Enterococcus casseliflavus*	KCTC 3552
	*Enterococcus cecorum*	KACC 13884
	*Enterococcus durans*	KCTC 13289
	*Enterococcus faecalis*	KCTC 5290
	*Enterococcus faecalis*	KACC 11859
	*Enterococcus faecalis*	KCTC 3206
	*Enterococcus faecium*	KACC 15681
	*Enterococcus faecium*	KACC 11954
	*Enterococcus faecium*	KCTC 13225
	*Enterococcus faecium*	KACC 14552
	*Enterococcus gallinarum*	NCCP 11518
	*Enterococcus gilvus*	KACC 13847
	*Enterococcus hirae*	KACC 16328
	*Enterococcus hirae*	KACC 10782
	*Enterococcus hirae*	KACC 10779
	*Enterococcus malodoratus*	KACC 13883
	*Enterococcus mundtii*	KCTC 3630
	*Enterococcus raffinosus*	KACC 13782
	*Enterococcus saccharolyticus*	KACC 10783
	*Enterococcus thailandicus*	KCTC 13134
	*Escherichia coli*	KCTC 1682
	*Escherichia coli*	ATCC 25922
	*Escherichia coli*	ATCC 23763
	*Escherichia coli*	ATCC 35150
	*Escherichia coli*	ATCC 43890
	Enteroaggregative *Escherichia coli*	NCCP 14039
	Enterohemorrhagic *Escherichia coli*	NCCP 11076
	Enteroinvasive *Escherichia coli*	NCCP 15663
	Enteropathogenic *Escherichia coli*	NCCP 13715
	Enterotoxigenic *Escherichia coli*	NCCP 15732
	*Listeria ivanovii*	ATCC 19119
	*Listeria monocytogenes*	ATCC 19115
	*Listeria monocytogenes*	KCTC 3569
	*Proteus mirabilis*	KCTC 2566
	*Proteus vulgaris*	KCTC 2579
	*Pseudomonas aeruginosa*	KCTC 1636
	*Pseudomonas chlororaphis*	KCCM 41854
	*Pseudomonas oryzihabitans*	KCCM 42984
	*Salmonella bongori*	ATCC 43975
	*Salmonella enterica* subsp. *arizonae*	ATCC 13314
	*Salmonella enterica* subsp. *diarizonae*	ATCC 43973
	*Salmonella enterica* subsp. *enterica*	ATCC 19585
	*Salmonella* Choleraesuis	ATCC 13312
	*Salmonella* Gallinarum	ATCC 9120
	*Salmonella* Paratyphi B	ATCC 10719
	*Salmonella* Paratyphi C	ATCC 13428
	*Salmonella* Typhimurium	ATCC 14028
	*Salmonella enterica* subsp. *houtenae*	ATCC 43974
	*Salmonella enterica* subsp. *indica*	ATCC 43976
	*Salmonella enterica* subsp. *salamae*	ATCC 15793
	*Shigella dysenteriae*	ATCC 13313
	*Shigella sonnei*	KCTC 2518
	*Vibrio cholerae*	NCCP 13589
	*Vibrio cholerae*	ATCC 14033
	*Vibrio cholerae*	ATCC 14035
	*Vibrio parahaemolyticus*	ATCC 17802
	*Vibrio parahaemolyticus*	KCCM 41664
	*Vibrio parahaemolyticus*	ATCC 27969
	*Vibrio vulnificus*	ATCC 33814

**Table 2 foods-10-02839-t002:** Specific primer information.

Species	Primer	Sequence	Size (bp)
*S. aureus*	Aureus_F	CAA GCA CAA GGC AGT GGT AT	145
	Aureus_R	GTG GCG TTG CAA TCT CCT TA	
*S. capitis*	Capitis_F	CGC AAG GTG GTC AAC TTG AT	150
	Capitis_R	GCG CAT CGT GAA GTA ATT CC	
*S. caprae*	Caprae_F	TCG TCG CAA CGA AGT TCA TC	146
	Caprae_R	CCT GGC GCA TAT GTA TGC TT	
*S. epidermidis*	Epidermidis_F	TGG CAC GGC TGG TAT TAG AG	121
	Epidermidis_R	GAC AGG ATG CGC GAT ACT TG	

**Table 3 foods-10-02839-t003:** Identification of strains isolated from food samples.

Sample Type	No. of Isolates	No. of Positive Results by Real-Time PCR			
		*S. Aureus*	*S. Capitis*	*S. Caprae*	*S. Epidermidis*
chicken (n = 4)	12	12	0	0	0
beef (n = 3)	11	11	0	0	0
pork (n = 5)	38	7	0	0	31
fish (n = 3)	6	6	0	0	0
salted fish (n = 5)	22	0	4	0	18
raw milk (n = 2)	14	0	0	0	14
Total (n = 22)	103	36	4	0	63

**Table 4 foods-10-02839-t004:** Identification of *Staphylococcus* contamination in food samples using the real-time PCR method developed in this study.

Sample Type	No. of Samples	No. of Positive Results by Real-Time PCR			
		*S. Aureus*	*S. Capitis*	*S. Caprae*	*S. Epidermidis*
beef	2	0	0	0	0
pork	8	1	0	0	0
lettuce	1	0	0	0	0
cucumber	1	0	0	0	0
raw milk	34	10	2	0	9
fermented fish	4	0	1	0	0
Total	50	11	3	0	9

## Data Availability

The data presented in this study are available on request from the corresponding author.

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
