# Peer review of "Real-Time PCR Method for the Rapid Detection and Quantification of Pathogenic Staphylococcus Species Based on Novel Molecular Target Genes"

_foods, 2021, doi:10.3390/foods10112839_

Round 1

Reviewer 1 Report

The manuscript titled ‘Real-time PCR method for rapid detection and quantification of pathogenic Staphylococcus species based on novel molecular target genes’ deals with the development of a real-time PCR method for the detection of four Staphylococcus species, S. aureus, S. capitis, S. caprae, and S. epidermidis, in order to be used for rapid monitoring of such bacteria in foods. The manuscript is well-written and gives a useful tool for rapid monitoring of such bacteria.

Only some minor comments

L 151 ‘Twenty-two (22)’

L 154 Twenty-five (25)

L 156 A volume of 0.1 ml /a quantity of…please check the manuscript for such issues

Author Response

Only some minor comments

L 151 ‘Twenty-two (22)’

Response: As you recommended, we revised the sentence in line 164 as follows:

Line 164: Twenty-two samples

L 154 Twenty-five (25)

Response: As you recommended, we revised the sentence in line 166 as follows:

Line 166: Twenty-five grams of each sample was transferred to

L 156 A volume of 0.1 ml /a quantity of…please check the manuscript for such issues

Response: We changed ‘0.1 ml of homogenized samples’ to ‘serial dilutions of homogenized samples’. As you recommended, we revised the issue such as ‘a volume of 0.1 ml /a quantity’ in the entire manuscript.

Line 168: Serial dilutions of homogenized samples

Line 138: 10 pmol/µl

Line 138: 20 ng/µl

Reviewer 2 Report

The paper entitled "Real-Time PCR method for rapid detection and quantification of pathogenic Staphylococcus species based on novel Molecular Target genes" deals with an interesting research topic. Altough in literature there are numerous protocols to identify coagulase positive or negative staphylococci, new protocols for their detection and quantification are welcome because of the importance of these protocols in the diagnosis, prophylaxis and prevention of staphylococcal toxaemias and for identifying and tracing staphylococci in food matrices.

I suggest to authors some minor revisions.

Staphylococcus aureus is a coagulase positive staphylococcus. The authors should pay more attention writing this. Please check and revise the abstract, lines 10-12 and the introduction, lines 33-34.

Lines 14-15, please efface this sentence from the abstract.

In the introduction, the authors should better highlight the issues of today existing protocols for the quantification and identification of staphylococci.

Figure 1 is a low quality figure, please replace the figure with a better quality one.

I checked and verified the specificity of the primers pairs with primer blast (https://www.ncbi.nlm.nih.gov/tools/primer-blast/). For S. aureus specific primers, I found that certain sequences (which I report below ) give an amplicon of 145 pb, please check if these sequences belong to S.aureus species:

(CP076027.1) Staphylococcus sp. MZ7 chromosome, complete genome

(CP076025.1) Staphylococcus sp. MZ1 chromosome, complete genome

(CP076029.1) Staphylococcus sp. MZ9 chromosome, complete genome

(CP076026.1) Staphylococcus sp. MZ3 chromosome, complete genome

(CP076028.1) Staphylococcus sp. MZ8 chromosome, complete genome

Author Response

  1. Staphylococcus aureus is a coagulase positive staphylococcus. The authors should pay more attention writing this. Please check and revise the abstract, lines 10-12 and the introduction, lines 33-34.

Response: As you recommended, we revised the sentence in lines 10 and 31-32 as follows:

Line 10: Coagulase-positive Staphylococcus aureus is a foodborne pathogen

Lines 31-32: Unlike foodborne pathogenic S. aureus, which is coagulase-positive staphylococci, S. capitis, S. caprae, and S. epidermidis are coagulase-negative staphylococci (CoNS)

  1. Lines 14-15, please efface this sentence from the abstract.

Response: As you recommended, we removed the sentence in lines 14-15 from the abstract.

  1. In the introduction, the authors should better highlight the issues of today existing protocols for the quantification and identification of staphylococci.

Response: As you recommended, we added the sentence on the issues of today existing protocols for quantification and identification of staphylococci in lines 52-62 and 70-71 as follows:

Lines 52-62: Various staphylococci detection methods exist, both phenotypic and genotypic. Other detection methods, such as the Staph-Zym test, API Staph test, and BF Phenix Automated Microbiology system, have been used for the detection of Staphylococcus species based on their phenotypic characteristics [10,11]. Recently, several attempts have been made to detect Staphylococcus species using matrix-assisted laser desorption ionization time-of-flight mass spectrometry based on protein expression profiles [12]. However, a low accuracy (50-70%) of biochemical reactions in the API systems and high initial acquisition cost for automatic mass spectrometry systems restrict their application [13,14]. Compared to genotypic detection methods, such as amplified fragment length polymorphic fingerprinting and PCR-based methods, phenotypic tests are less accurate [1,14,15].

Lines 70-71: At present, real-time PCR is used for monitoring and detection of Staphylococcus species in food processing [1,22,23].

  1. Figure 1 is a low quality figure, please replace the figure with a better quality one.

Response: As you recommended, we replaced figure 1 with a better quality one (resolution of the figure to 300 dpi).

  1. I checked and verified the specificity of the primers pairs with primer blast (https://www.ncbi.nlm.nih.gov/tools/primer-blast/). For S. aureus specific primers, I found that certain sequences (which I report below) give an amplicon of 145 pb, please check if these sequences belong to S. aureus species:
    (CP076027.1) Staphylococcus sp. MZ7 chromosome, complete genome
    (CP076025.1) Staphylococcus sp. MZ1 chromosome, complete genome
    (CP076029.1) Staphylococcus sp. MZ9 chromosome, complete genome
    (CP076026.1) Staphylococcus sp. MZ3 chromosome, complete genome
    (CP076028.1) Staphylococcus sp. MZ8 chromosome, complete genome

Response: As you recommended, we checked if these five genome sequences belong to S. aureus species using average nucleotide identity (ANI) analysis. Also, we newly checked the specificity of four primer pairs and added the sentence in lines 123-124, 230-236, and 378-379 as follows:

Lines 123-124: The specificity of designed primer pairs was checked using the primer-BLAST tool [31].

Lines 230-236: Species-specific primer pairs were designed from each molecular target gene, and the specificity was confirmed through in silico analysis. As a result, all primer pairs produced amplicons only in the target species (Table S3). For S. aureus specific primers, we found that certain genome sequences, including Staphylococcus species MZ1, MZ3, MZ7, MZ8, MZ9, T93, SM3655, and SM9054, give an amplicon of 145 bp. Eight genomes of Staphylococcus species showed 98.83 to 99.87% identities with S. aureus type strain (DSM 20231), suggesting that these genomes belong to S. aureus.

Lines 378-379: Table S3: Specificity of designed primer pairs using primer-BLAST tool.

Table S3: We newly added Table S3 for the specificity results of the designed primer pairs.

Reviewer 3 Report

Dear authors,

In this manuscript a real-time PCR method for detecting Staphylococcus species is developed. Members of this genus can cause food poisoning mostly through intoxication by numerous enterotoxins that the bacteria produce however for most consumers the illness usually completes its course within a day or two. Generally, S. aureus must grow to populations of >106 CFU per gram to produce sufficient amounts of enterotoxin to produce the illness.

New techniques as WGS can provide new information which is nicely used in this study to identify new targets for use to identify members of this genus.

Introduction: Reference number 1 (an article published in 2022) is used as a reference to very general aspects of this pathogen and the disease. The authors should find a more appropriate reference for these general considerations.

Line 49—53 Methods for detecting these genus’ is only generally described (detecting pathogens). More detailed descriptions are necessary due to the objective of this manuscript. Additionally, real time PCR have earlier been described for this genus.  

Materials and methods: OK

Results: It is difficult to read and understand Figure 1. Two figures? Legends do not explain to the reader what actually can be seen in the figure. More details are required.

It would be interesting to know the identity of the target genes, and then compare and discuss related to earlier results from other authors (line 64-67).

The analysis of food samples is based on isolating the bacteria from the food samples first. This is a time-consuming method (line 52), and the advantage of using real-time PCR for a quick detection does not exist.   Have the authors tried to analyze DNA directly from the food sample and thereby identify the different species present in the food product? That would be adequate to situation in a food business where you want to screen your product for any contamination.

Discussion: Most of the discussion is relevant but general comments (line 291-294) could be more specific if compared to earlier studies within development of  real time PCR assays.

Author Response

  1. Introduction: Reference number 1 (an article published in 2022) is used as a reference to very general aspects of this pathogen and the disease. The authors should find a more appropriate reference for these general considerations.

Response: As you recommended, we added more appropriate reference (Argudín et al., 2010, Toxins, doi: 10.3390/toxins2071751) in lines 28-31 as follows:

Lines 28-31: Staphylococcus aureus is one of the most important foodborne pathogens, producing staphylococcal enterotoxins causing diarrhea and vomiting through direct invasion or systemic transmission, adversely affecting human health [1,2]

  1. Line 49-53 Methods for detecting these genus’ is only generally described (detecting pathogens). More detailed descriptions are necessary due to the objective of this manuscript. Additionally, real time PCR have earlier been described for this genus.

Response: As you recommended, we added more detailed descriptions for methods for detecting Staphylococcus genus in lines 52-62 and 70-71 as follows:

Lines 52-62: Various staphylococci detection methods exist, both phenotypic and genotypic. Other detection methods, such as the Staph-Zym test, API Staph test, and BF Phenix Automated Microbiology system, have been used for the detection of Staphylococcus species based on their phenotypic characteristics [10,11]. Recently, several attempts have been made to detect Staphylococcus species using matrix-assisted laser desorption ionization time-of-flight mass spectrometry based on protein expression profiles [12]. However, a low accuracy (50-70%) of biochemical reactions in the API systems and high initial acquisition cost for automatic mass spectrometry systems restrict their application [13,14]. Compared to genotypic detection methods, such as amplified fragment length polymorphic fingerprinting and PCR-based methods, phenotypic tests are less accurate [1,14,15].

Lines 70-71: At present, real-time PCR is used for monitoring and detection of Staphylococcus species in food processing [1,22,23].

  1. Materials and methods: OK

  1. Results: It is difficult to read and understand Figure 1. Two figures? Legends do not explain to the reader what actually can be seen in the figure. More details are required.

Response: Figure 1 is difficult to separate into two figures. As you recommended, we revised a legend to explain to the reader more details in lines 200-204 as follows:

Lines 200-204: Pan-genome analysis generated with Anvi’o software (version 7.0) for 155 staphylococci genomes. The layers represent individual staphylococci genomes organized by their phylogenomic relationship. In the layers, the dark and bright colored areas within the bar represent the presence and absence of genes, respectively. The ANI values are represented by heatmap determined at high (black) and low (gray) similarities.

  1. It would be interesting to know the identity of the target genes, and then compare and discuss related to earlier results from other authors (line 64-67).

Response: As you recommended, we newly compared target genes found by other authors and added the sentence in lines 116-118, 222-229, 328-331, and 377-378 as follows:

Lines 116-118: The specificity of discovered molecular target genes with the reported target genes (tuf, sodA, nuc, and dnaJ) was confirmed by aligning them with 94 S. aureus, S. capitis, S. caprae, and S. epidermidis genomes.

Lines 222-229: We compared the specificity of the newly discovered molecular target genes with the reported target genes in detecting four Staphylococcus species and found a better specificity for newly discovered molecular target genes in this study (Table S2). Novel molecular target genes showed 100% presence in the target species while partial presence in the target species for the reported genes (tuf, sodA, nuc, and dnaJ). All target genes except the nuc gene were present with high similarities (86%-100%) not only to the target species but also to the non-target species. These results indicated that the four molecular target genes discovered in this study are suitable for the detection of four Staphylococcus species.

Lines 328-331: However, a previous study has reported that the 16S rRNA gene of S. epidermidis, S. capitis, and S. caprae share high sequence similarities (≥97%) without exhibiting sufficient variabilities to allow for differentiation between species [39].

Lines 377-378: Table S2: Presence of novel and reported molecular target genes for target and non-target genomes

Table S2: We newly added Table S2 for result of comparing the specificity of newly discovered molecular target gene with the reported target genes.

  1. The analysis of food samples is based on isolating the bacteria from the food samples first. This is a time-consuming method (line 52), and the advantage of using real-time PCR for a quick detection does not exist. Have the authors tried to analyze DNA directly from the food sample and thereby identify the different species present in the food product? That would be adequate to situation in a food business where you want to screen your product for any contamination.

Response: As you recommended, we newly tried to analyze DNA directly from the food sample and thereby identify the different species present in the food product. We added the sentence in lines 172-180 and 302-312 as follows:

Lines 172-180: 2.9. Detection of Staphylococcus species in food samples
Fifty samples including ten samples of meat (beef, n = 2; pork, n = 8), two samples of ready-to-eat vegetables (lettuce, n = 1; cucumber, n = 1), 34 samples of raw milk, and four samples of salted fish, were randomly collected from local markets in Korea. To extract genomic DNA from food samples, 10 g of sample was homogenized using a blender (model: Tokebi Origin, Buwon Electronics, Seoul, Korea). Genomic DNA of food samples was extracted under the conditions described in section 2.4. Detection of four Staphylococcus species in unknown samples was performed under the conditions described in section 2.5.

Lines 302-312: 3.6. Detection of four Staphylococcus species contamination in food samples
To confirm the feasibility of real-time PCR for the detection of Staphylococcus species contamination in actual samples, 50 samples were tested using the real-time PCR method developed in this study. S. aureus was detected in 11 samples of pork and raw milk, and S. epidermidis was detected in 9 samples of raw milk. S. capitis was detected in three samples of fermented fish and raw milk. S. caprae was not present in any of the food samples. The test result showed that the detection rates of Staphylococcus species were 22% for S. aureus, 6% for S. capitis, and 18% for S. epidermidis (Table 4). These results are consistent with a previous study suggesting that S. epidermidis and S. aureus are the main staphylococcal contamination in food, possibly caused by staphylococcal infection during the storage and sale of related foods [1].

Table 4: We newly added Table 4 for result of analysis of DNA directly from the food samples.

  1. Discussion: Most of the discussion is relevant but general comments (line 291-294) could be more specific if compared to earlier studies within development of real time PCR assays.

Response: As you recommended, we added the sentence in lines 347-358 as follows:

Lines 347-358: Real-time PCR method provides a tool for sensitive and accurate quantification of target bacteria that could be applied to detect Staphylococcus species in foods [22,23,44]. Although several real-time PCR methods for detecting Staphylococcus species have been reported, their target genes or sequences have been found to exhibit poor specificity and produce false-positive results [1]. Recently, a real-time PCR method targeting novel specific genes obtained by pangenome analysis for efficient and accurate detection of pathogenic Staphylococcus species has been developed [1]. This method displayed a better specificity than the previous real-time PCR method. However, for monitoring of pathogenic Staphylococcus species using existing real-time PCR methods, researchers focused on mining specific genes for S. aureus and S. epidermidis [1], while no S. capitis and S. caprae, which are closely related species to S. epidermidis, target gens have been reported.

Round 2

Reviewer 3 Report

Dear authors

Success with your manuscript